# Identification of Important Factors Affecting Use of Digital Individualised Coaching and Treatment of Type 2 Diabetes in General Practice: A Qualitative Feasibility Study

**DOI:** 10.3390/ijerph18083924

**Published:** 2021-04-08

**Authors:** Pernille Ravn Jakobsen, Jeanette Reffstrup Christensen, Jesper Bo Nielsen, Jens Søndergaard, Dorte Ejg Jarbøl, Michael Hecht Olsen, Jens Steen Nielsen, Jette Kolding Kristensen, Carl J. Brandt

**Affiliations:** 1Research Unit of General Practice, Department of Public Health, University of Southern Denmark, 5000 Odense C, Denmark; jrchristensen@health.sdu.dk (J.R.C.); JBNielsen@health.sdu.dk (J.B.N.); jsoendergaard@health.sdu.dk (J.S.); DJarbol@health.sdu.dk (D.E.J.); cbrandt@health.sdu.dk (C.J.B.); 2Research Unit of User Perspectives and Community-Based Interventions, Department of Public Health, University of Southern Denmark, 5000 Odense C, Denmark; 3Steno Diabetes Center, Odense University Hospital, 5000 Odense C, Denmark; Jens.Steen.Nielsen@rsyd.dk; 4Department of Internal Medicine and Steno Diabetes Center Zealand, Holbaek Hospital, 4300 Holbaek, Denmark; michael.olsen@dadlnet.dk; 5Department for Regional Health Research, University of Southern Denmark, 5000 Odense C, Denmark; 6Danish Center for Strategic Research in Type 2 Diabetes, Steno Diabetes Center Odense, Odense University Hospital, 5000 Odense C, Denmark; 7Center for General Practice, Aalborg University, 9220 Aalborg, Denmark; jkk@dcm.aau.dk; 8Liva Healthcare, 1434 Copenhagen, Denmark

**Keywords:** type 2 diabetes, general practice, eHealth, lifestyle, qualitative, feasibility study

## Abstract

Most type 2 diabetes patients are treated in general practice and there is a need of developing and implementing efficient lifestyle interventions. eHealth interventions have shown to be effective in promoting a healthy lifestyle. The purpose of this study was to test the feasibility, including the identification of factors of importance, when offering digital lifestyle coaching to type 2 diabetes patients in general practice. We conducted a qualitative feasibility study with focus group interviews in four general practices. We identified two overall themes and four subthemes: (1) the distribution of roles and lifestyle interventions in general practice (subthemes: external and internal distribution of roles) and (2) the pros and cons for digital lifestyle interventions in general practice (subthemes: access to real life data and change in daily routines). We conclude that for digital lifestyle coaching to be feasible in a general practice setting, it was of great importance that the general practitioners and practice nurses knew the role and content of the intervention. In general, there was a positive attitude in the general practice setting towards referring type 2 diabetes patients to digital lifestyle intervention if it was easy to refer the patients and if easily understandable and accessible feedback was implemented into the electronic health record. It was important that the digital lifestyle intervention was flexible and offered healthcare providers in general practice an opportunity to follow the type 2 diabetes patient closely.

## 1. Introduction

Type 2 diabetes (T2D) has an increasing prevalence globally, and there is substantial potential to improve prevention as well as treatment, thereby reducing societal costs and increasing patients’ quality of life [1]. Successful treatment to reduce the risk factors and prevent complications of T2D typically comprises many components, including medicine and lifestyle changes [2].

In Denmark there are around 300,000 adult diabetes patients. Approximately 90% have T2D, with the majority (80%) being treated in primary health care [3]. In 2016, the overall prevalence was 5.5% for T2D [4]. According to Danish clinical guidelines, patients should be supported in self-management of the disease, healthy lifestyle, and adherence to medication at consultations. Therefore, primary care is an essential part of the treatment paradigm for individuals diagnosed with T2D [5].

Despite great societal efforts and a high focus on T2D in general practice, many patients are not treated optimally, nor do they sufficiently follow the recommendations of healthy lifestyle [6]. Studies have shown that the consultations in general practice frequently do not address lifestyle issues [7]. Linmans et al. found that T2D patients are often passive during the consultation. They concluded that T2D patients must be stimulated to take an active role in their treatment and that patients and their general practitioner (GP) or practice nurse (PN) should become equal partners in determining mutually agreeable treatment plans and goals [8].

The main challenge faced by many researchers in T2D treatment and lifestyle interventions is insufficient self-management of the disease [9]. Studies have found that education in diabetes self-management can improve glycemic control and potentially postpone and/or prevent disease complications. Hence, patients newly diagnosed with T2D are often referred to face-to-face group courses [10,11,12]. However, studies suggest that the face-to-face courses for some are difficult to attend because of the timing of the courses, the lack of transport, work and family commitments, or they do not like to participate in groups [13]. Internet- and app-based self-management interventions bypass some of these barriers and are suggested as an alternative way to improve lifestyle behaviours among patients with T2D [14]. Despite the fact that online diabetes self-management interventions are promising, there are, however, also challenges including low uptake and engagement that can limit effectiveness and need further research [15].

In 2011 Vigersky stated that technology-driven innovative solutions are desperately needed to assist both patients and their healthcare providers in overcoming the challenges of T2D [16]. Now ten years later, studies have shown that eHealth interventions can be effective in promoting healthy behaviour, which may be further improved by integrating self-regulation techniques (e.g., personal goal settings and coping strategies). In doing so, eHealth may cover some of the tasks of GPs prompting them to address self-management in the consultation [17]. The most promising eHealth tools and techniques in T2D management comprise regular collection of biometric data through devices (e.g., glucometers, activity and step monitors) [18,19], SMS/text messaging [18,20,21], secure email communication with clinical teams, and regular reporting of quality-of-life variables [22,23].

In the light of this an eHealth lifestyle coaching intervention has been developed based on the experiences from previously developed eHealth solutions on which extensive research has been conducted [24,25,26,27]. The eHealth lifestyle coaching intervention was tested and implemented in eight Danish municipalities in 2016–2018, and the effects of the eHealth intervention among T2D patients were recently assessed [28]. The findings showed that the eHealth intervention significantly helped reduce weight among T2D patients, and the authors concluded that there is a positive effect of real-life eHealth intervention on T2D patients’ lifestyle in a municipal setting [28].

Given the challenges and the potential of eHealth lifestyle interventions within T2D, we aim to test the feasibility of a newly developed digital individualised coaching and lifestyle treatment intervention of T2D in general practice (DICTA). The digital lifestyle treatment intervention we test in this study uses the same lifestyle app as the intervention tested in municipal settings mentioned above [28,29]. However, in the general practice setting, the first coaching session takes place via video consultations by a trained external health coach and not in person by a municipality employed health coach such as in the municipalities. For additional information about the digital lifestyle intervention, please see Textbox 1 in the Appendix A.

The present study is the first phase of a randomised controlled trial (RCT)-study. Our specific aim with this first study was to describe attitudes and behaviour towards lifestyle interventions in general practice, as well as to evaluate the feasibility of the intervention by identifying factors of importance when conducting digital individualised and collaborative treatment of T2D in a general practice setting. The identified factors will subsequently be included in the development of the final digital intervention prior to conducting the RCT.

## 2. Materials and Methods

We conducted a qualitative feasibility study by performing focus group interviews in four general practices.

### 2.1. Recruitment and Participants

The authors are thoroughly versed in general practice. Hence, we had an overview of general practices and strategically selected general practices that were relevant to include. We aimed to include both GPs and PNs. To assure a certain variation in age, gender, and level of interest in T2D, lifestyle interventions and eHealth, we adopted a purposeful sampling strategy including GPs with high/low interest in T2D and lifestyle interventions, high/low interest in eHealth, GPs of different gender and age, and GPs with different practice types [30]. Four general practices in the Region of Southern Denmark were invited and agreed to participate in the feasibility study. The focus group interviews were conducted in each general practice partly to avoid transportation and time barriers [31] and partly to be able to show the prototype of the digital lifestyle intervention in DICTA on practice’s own computers.

### 2.2. Data Collection

Prior to the focus group interviews, an interview guide was developed in an iterative process by the research team as recommended by Kvale and Brinkmann [32]. The interview guide included general questions related to management of T2D patients and lifestyle interventions in general practice as well as the participants’ general experiences of collaboration with the municipality and their lifestyle interventions. Further, the interview guide included questions related to digital lifestyle coaching after the prototype of the DICTA intervention was introduced to the participants by the last author. Finally, the interview guide included questions related to incentive structures and important factors in digital lifestyle coaching of T2D patients in general practice. Prior to the focus group interviews the interview guide was tested and validated by colleagues at the research unit.

The focus group interviews were facilitated by the last author and supported by a research assistant. The interviews were conducted using open-ended questions supplemented with follow up questions to allow the participants to discuss freely. If there was a need for elaboration, the participants were asked exploratory and specifying questions as recommended by Kvale and Brinkmann when conducting focus group interviews [32].

### 2.3. Data Analysis

All focus group interviews were recorded and transcribed verbatim within 14 days after each interview. Data were subject to a thematic analysis by the research team, guided by Braun and Clarke’s five step framework [33]. Analysis was carried out inductively and involved researchers, as shown in Table 1. Saturation was reached since no new themes or codes were found during the last coding process.

The quotes that best illustrated the emerging themes were selected and translated from Danish to English by PRJ. All quotes were evaluated continuously by the co-authors, and changes were made to reach agreement prior to writing this paper.

## 3. Results

During August and September 2019, four general practices were recruited. In total, 15 health professionals participated in the focus group interviews. Six were PNs and nine were GPs. Eight were female and seven were male. All participating general practices were partnerships practices and half of them were in urban areas. For participant characteristics, see Table 2. The focus group interviews lasted 46–87 min with an average length of 63 min.

The analysis identified two overall themes and four associated subthemes, see Table 3. In the following, we will present the themes and subthemes describing factors of importance when conducting digital individualised and collaborative treatment of T2D in general practice.

### 3.1. Theme 1: Distribution of Roles and Lifestyle Interventions in General Practice

In the first part of the focus group interviews general questions related to management of T2D patients and lifestyle interventions in general practice as well as the participants’ general experiences of collaboration with the municipality and lifestyle interventions were discussed. This led to findings related to distribution of roles and lifestyle interventions in general practice.

#### 3.1.1. Subtheme: External Distribution of Roles

In general, all participants were aware of the existing diabetes courses in the municipalities where the aim is to support the patients in better self-management of their disease through lifestyle interventions. However, it varied whether the GPs referred their patients to the courses. Practice no. 1 and no. 3 used to refer the patients to diabetes courses in the municipalities but experienced varying degrees of course quality and response from the healthcare providers in the municipalities. It also varied whether the GPs or PNs followed up on the patients after they had completed the course to support the patients in continuing self-management of T2D and maintenance of a healthy lifestyle. In practice no.1, the GP and the PN discussed whether they received any feedback from the municipality after the referred patients had finished their diabetes course:

*“We do not receive* *any reporting from the municipality when we refer our patients to diabetes courses.”*(GP 1,1)


*“We do, I read it every time. The report describes that the patient has participated in a diabetes course with good effects, but well—The information is not that useful.”*
(PN 1,1)

Thus, external actions are dependent on internal behaviour and actions. Practice no. 2 reported that they have their own smoking cessation instructor and did in general not refer their patients to courses in the municipalities. Instead they supported their patients in lifestyle interventions in-house. Practice no. 4 sometimes referred their patients to courses in the municipalities. However, since they often experienced that the patients were not willing to participate, they have stopped referring the patients.

Practice no. 3 suggested that the diabetes courses could be more up to date and be accessible through a computer or as video consultations and eLearning, since they experienced that most of the existing courses in the municipality do not fit into the patients’ daily lives due to work or other daily activities. Furthermore, they, similar to practice no. 1, expressed that they often do not receive any usable feedback after the patients have finished their course. Hence, follow-ups on the achieved lifestyle goals are not prioritised during the consultations.

After the digital intervention was presented in practice no. 2, it raised some questions as to the distribution of roles between the digital health coach and the PNs in general practice.


*“If they talk to the coach about lifestyle issues one day, they do not need to come and talk to me about the same things the same day. I think we need to make an agreement about what they discuss with the coach and what they discuss with me.”*
(PN 2,1)

#### 3.1.2. Subtheme: Internal Distribution of Roles

When discussing roles in relation to lifestyle interventions in general practice, all practices had an internal agreement that the PNs most often had the contact to the municipalities. As GP 2 from practice no. 4 described:


*“We have an agreement that it is the PN’s responsibility to refer the patients to the courses and follow up on what is happening in the municipality.”*
(GP 4,2)

In general, the PNs played a major role in management of T2D in the participating practices. In practice no. 3, it was the PN who had signed up to the DICTA project because of a special interest in T2D. She was described as “the diabetes superstar” and was very motivated for new ways of supporting the patients in lifestyle interventions and self-management. There were differences between the practices, but also internally, in how much the GPs followed up on lifestyle interventions in the consultation. As discussed in practice no. 3:


*“I talk about diet, physical activity, smoking cessation and alcohol in overall terms. I do not go into details with counting steps and so.”*
(GP 3,1)


*“Well, I go a lot into details about steps. I experience that many of my patients are wearing a step counter. I think it’s an easy and concrete way of discussing their physical activity level.”*
(GP 3,2)

In the discussions, it was clear that the GPs were responsible for the biomedical aspects such as medication, and the PNs were responsible for the follow-up consultations and lifestyle guidance of the T2D patients. The PNs also raised the psychosocial challenges such as motivation of the patients to follow a healthier lifestyle. The PN in practice no. 2 talked about the need of having a digital tool that could be used for patients, GPs, and PNs to allow better support for the patients in lifestyle interventions.

### 3.2. Theme 2: Pros and Cons for Digital Lifestyle Interventions in General Practice

The prototype of the DICTA intervention was introduced to the participants in the focus group interviews. In the second part of the focus group interviews, the interview guide included questions related to digital lifestyle coaching. Additionally, questions related to incentive structures and important factors in digital lifestyle coaching of T2D patients in general practice were discussed. This led to findings related to pros and cons for digital lifestyle interventions in general practice.

#### 3.2.1. Subtheme: Access to Real Life Data

All participants in the four focus group interviews could see advantages of the digital health solution and the ability to gain access to real life data from the patients and how they manage their T2D in their daily life. By having the opportunity to gain access to real life data from the patients, the GPs and PNs agreed that it could be a way to set more realistic goals together with the patients. Both PNs and GPs shared their experiences from consultations where patients often promise to begin a healthier lifestyle, but when they met up three months later, nothing had happened:


*“I think this is really helpful to have this motivational talk with the patients. Because, we know our patients and we all know that they do not follow our recommendations when they get home. This eHealth tool will help us to better understand what the barriers and challenges are in adopting a healthier lifestyle. And if the patient succeeds taking more steps for instance, he will become so happy when he gets his results from his blood tests.”*
(PN 2,1)

There was generally a positive attitude towards the digital health solution, as described by one of the GPs:


*“I can imagine the gains, if we by using the real life data on my screen in the consultation can set up more realistic and concrete goals instead of just telling the patient to walk and cycle a little more. With this app you can say, now you take 1500 steps per day. How can we increase that to 7000?”*
(GP 1,1)

However, later in the focus group interviews, and after having discussed the pros of the digital solution, the participants began to discuss the cons. In particular, time is a barrier in general practice as described by the participants:


*“We only have 15 min when we see the patient, and in addition to look on the screen we also need to talk to them.”*
(GP 4,1)


*“I need to know what exactly I should say to the patient within 30 s, because I do not have more time than that to clarify and nuance the data shown on the screen.”*
(PN 1,1)


*“Well it opens for a lot of questions, and we do not have more time to the patients.”*
(PN 3,2)


*“I agree. We do not need more registration tasks, and I suspect a little extra time-consuming registration unless you become so familiar with this platform, so you get a quick overview very easily.”*
(GP 3,1)

#### 3.2.2. Subtheme: Change in Daily Routines

Despite being mostly positive towards the digital health solution, the participants discussed pros and cons for implementation of the digital solution in their daily routines. Many participants highlighted the importance of the technology being integrated as a part of the existing systems in general practice and that it should run perfectly as soon as it is integrated.


*“If this is going to work in daily practice, it is important that it is easily accessible and that it runs perfectly from the very beginning.”*
(GP 4,1)

GP 1 from practice no. 2 emphasised that easy access and usability was very important, if he should use it:


*“There need to be an icon that works. It should only take one extra click to get the data, and it should be easy to get an overview. Then I would use it when the patient visits me in the motivational conversation with the patient and to point out the good things.”*
(GP 2,1)

The participants expressed differing views on whether the digital solution would lead to fewer or more consultations. In practice no. 3, the PN expressed forcefully:


*“If this requires more consultations, it means no thanks from me.”*
(PN 3,2)

She elaborated:


*“We have to be aware about the number of consultations per year since we get a fixed fee per year per T2D patient.”*
(PN 3,2)

The facilitator asked whether she would be more motivated for the digital solution if it could help her decrease the number of consultation and she answered “yes!”. In addition, it would motivate her if the consultation could be less time consuming.

Fewer consultations were also mentioned by GP 2 in practice no. 3. He expressed that it could be a reward for the patients if they were compliant and registered their daily activities to reach their goals, and the GP could follow the patients’ development on his own screen and just write to the patients that they were doing good and hence do not need to meet up in the consultation.

In practice no. 2, the discussions in the focus group interview turned out differently. The participants were in general very positive towards the digital solution and opportunities to have access to real life data from the patients. However, they pointed to the need of making some agreements with the patients on how often they would look into the registered data:


*“We will enter into an agreement with the patients about when we will look at their data, because we do not follow up on them every day.”*
(PN 2,1)

In practice no. 2, the participants discussed a possible increase in consultations since the digital solutions contribute to more interesting content than only feedback on blood tests and biomedical investigations.


*“I think that this will make me feel that both I and the patient are getting more out of the consultation. Our routine practice is that we see the patients every third month, but if we are getting more real-life data to work with it will be more interesting to see the patient one time per month.”*
(GP 2,1)

The identified themes and associated subthemes in relation to attitudes and behaviour towards digital lifestyle interventions contain different important factors affecting use of digital individualised coaching of T2D patients in a general practice setting. Table 4 sums up the identified factors of importance when it comes to adopting digital lifestyle coaching in a general practice setting.

## 4. Discussion

We found that it is important to be aware of the distribution of roles externally and internally when it comes to optimising the treatment of T2D through digital lifestyle interventions in a general practice setting. We also found that both GPs and PNs expressed pros and cons for offering digital lifestyle interventions to T2D patients in general practice. In the following, the factors of importance will be elaborated and discussed in relation to the current literature.

The vast majority of T2D patients in the Scandinavian countries are diagnosed and managed in primary care. Often, it is the only treatment institution that many patients encounter [34]. Patients in Denmark are listed with the same GP usually for many years, which allow for creating trust and a close patient–doctor relationship, and thus this translates into improved cooperation in the process of prevention and treatment [35]. Studies have shown that GPs consider health promotion and chronic disease prevention as important aspects of their work [36]. However, GPs notice obstacles in their implementation such as lack of time and insufficient financial support [37]. Furthermore, there is a growing concern among GPs that preventive lifestyle initiatives may not provide the expected benefits, and therefore GPs do not pay that much attention to lifestyle interventions in the consultation [38]. We found similar opinions in our study, and even though lifestyle intervention is a common task in primary care, most of the participating practices referred T2D patients to external healthcare providers mostly based in the municipalities. As expressed by the participating GPs and PNs, there are some challenges in the present set- up in the context where this study took place. Based on their experiences there might be a need for better communication and collaboration with external lifestyle interventions providers. In addition, a more continuous feedback and a report when the patient has finalised the lifestyle intervention in the municipalities is needed. Otherwise, the participating GPs and PNs found it difficult to follow up on the lifestyle changes that the patient has achieved and thereby support the patient in continuing self-management of T2D in the routine controls.

A recently published Australian study reported that barriers to refer patients to digital interventions were among others the lack of knowledge of effective apps, the lack of a trustworthy source to access them, and the lack of understanding of benefits [39]. Our study identified similar important factors that could be a barrier if GPs and PNs should refer the patients to a digital lifestyle intervention. We found that it is important to inform general practices about the digital lifestyle intervention as a valid offer that they can refer their T2D patients to. Additionally, we found that it is important that the GPs and PNs understand the content and the benefits of the digital coaching sessions and that it is provided by an external healthcare provider they can trust. In addition, it is important that the distribution of roles is clear so that the patient experience a collaboration between the healthcare providers in primary care and external healthcare providers.

In the focus group interviews, we found that the internal distribution of roles was characterised by GPs having the responsibility of correct medication and PNs having the responsibility of routine follow-up visits and support in self-management of the disease and lifestyle interventions. In a Dutch study, they found that T2D patients preferred to have lifestyle consultations with PNs, as the PNs set aside more time for the appointment. Further, they found that the T2D patients often have a trusting relationship with the PNs, which is an important factor in order to involve and motivate the patients [7]. This indicates the importance of being aware of the existing internal distribution of roles when introducing a new lifestyle intervention since both GPs and PNs are involved in the treatment of T2D patients in Denmark.

Studies have shown that T2D patients often are not motivated to increase their healthy lifestyle and that many patients have a low physical activity level. Furthermore, the authors claim that healthcare professionals often do not know whether patients are motivated to change their lifestyle and should therefore assess motivation regularly to optimise lifestyle management [40]. This issue was discussed in our focus group interviews and especially the pros of having a digital lifestyle intervention that could provide both GPs and PNs with real life data on the patients’ compliance. Having the opportunity to follow up on how the individual T2D patients perform in their daily lives by having real life data was experienced as a great advantage of the digital lifestyle intervention. However, it was important that it was not time consuming, and it should be easy to obtain a quick overview of the data during the consultation.

Shortage of time is well known as a barrier in primary care. Hence it was raised as an important factor that the digital lifestyle intervention should support both GPs and PNs in their daily practice and not require more time or more visits. Some participating GPs and PNs claimed that it could be motivating if the digital solution could decrease the number of visits. However, another GP expressed that it could be more interesting to have more visits when the patients started to use the digital lifestyle intervention. Brotons et al. claim that in a patient-centered approach, patients become important partners in medical care. In addition, for preventive interventions in general practice to succeed, patients’ points of view must be considered in addition to those of GPs [41]. Since both GPs and PNs expressed different points of view, it underlines the importance of a flexible solution that offers different kinds of engagements from the GPs and PNs. GPs and PNs with a special interest in T2D and lifestyle intervention might want to follow up on their patients more closely, while other GPs and PNs might expect the digital solution to facilitate fewer consultations and more efficient visits. These are important factors to bear in mind when testing a digital lifestyle intervention in general practice and underline the importance of engaging healthcare providers from general practice in the development of the digital lifestyle intervention.

### 4.1. Strengths and Limitations

The use of qualitative methods in this study allowed in-depth understandings of GPs’ and PNs’ experiences and opinions on dealing with T2D patients to better understand their needs. This is important prior to developing and testing a digital intervention in a healthcare setting [42]. We chose focus group interviews rather than individual interviews as a focus group interview is appropriate when the aim of the study is to explore different perceptions of a given phenomenon. Furthermore, the interaction within a focus group is important and leads to discussion of the participants’ different experiences and perceptions [31]. In this study, the focus group interviews were conducted using open-ended questions and some follow-up questions to allow the participants to discuss freely. The study included GPs and PNs recruited to the study through the authors’ network, meaning that barriers of importance in relation to digital lifestyle interventions might not be articulated sufficiently. However, our data showed that the participants expressed both negative and positive opinions to the digital lifestyle intervention in the DICTA study. Furthermore, since digital lifestyle intervention is a relatively new term in a general practice setting, it might be a challenge for the participants to have the full understanding of the intervention, and hence informants may express a combination of expectations and perceptions.

A multidisciplinary research group conducted the study (physiotherapist, occupational therapist, GPs, and hospital physicians specialised in T2D). Multidisciplinary and continuous discussions throughout the whole research process, from development of the interview guide to analysis and preparation of the paper, helped minimise the risk of our findings being dominated by preconceptions. In the next phases of the study, more researchers will be involved, including data science specialists.

### 4.2. Practical Implications and Recommendations for Further Research

Digital health and, in particular, health apps hold a great potential for improving self-management and thereby health outcomes in patients with chronic conditions such as T2D. However, as we have identified in this study, some important factors are critical to adoption in general practice. We identified different needs that should be taken into consideration when testing digital health interventions in general practice. In particular, it is important that the GP and PN know the content of the eHealth solution since they are key persons in referring patients to external providers of healthy lifestyle interventions. Furthermore, it is important that if they refer patients to a lifestyle intervention, they will receive easily understandable feedback from the external provider that is integrated directly into the electronic health record and gain easy access to real-life data to be able to support the patients in self-management of their chronic disease when the patients visit their GP or PN. Both PNs and GPs are aware that time is a scarce resource in daily practice. Hence, an eHealth solution should lead to less time-consuming and more effective consultations.

We identified a lack of digital culture in general practice, and both GPs and PNs seem to feel that eHealth lifestyle interventions are inconvenient. In addition, there is a lack of clear distribution and understanding of roles when it comes to digital lifestyle interventions in general practice.

In a newly published paper, Gordon et al. claim that most health apps do not deliver value, especially not for patients with chronic conditions, and if they are to be effective, both patients and providers need to gain value from utilising these tools. They also claim that if apps could be “prescribed” to patients through existing workflows, patients and clinicians may be more likely to use them. They suggest that apps should be integrated with clinical decision support systems to ensure appropriateness [43]. Hence, further research should focus on how digital lifestyle coaching could support clinical decision making to optimise the treatment of T2D in general practice. This will be the topic for future studies.

## 5. Conclusions

Our specific aim with this study was to evaluate the feasibility of a digital lifestyle intervention by identifying factors of importance when conducting digital individualised and collaborative treatment of T2D in a general practice setting.

If the digital lifestyle intervention should be feasible it is of great importance that the GPs and PNs know the role and content of the intervention. In our study, the GPs and PNs were positive towards referring T2D patients to a digital lifestyle intervention if it was easy to refer the patients and if they received easily understandable and accessible feedback that was implemented directly into the electronic health record.

The digital lifestyle intervention should support a good collaboration between the GP, the PN, the patient and the coach in the digital lifestyle intervention. Thus, it is important that the digital lifestyle intervention is flexible and offers healthcare providers in general practice an opportunity to follow the T2D patient closely through what is registered in their smartphone and wearable. However, it should not be mandatory since most GPs and PNs are concerned that they do not have the time or motivation to pay more attention to lifestyle issues.

The identified factors will subsequently be included in the development of the final digital intervention prior to conducting the RCT.

## Figures and Tables

**Table 1 ijerph-18-03924-t001:** Thematic analysis.

Step	Description
I	Familiarisation with data (PRJ, CJB)
II	Inductive coding of data (PRJ and CJB coded data independently and generated an initial list of ideas about what is in the data)
III	Discussion of codes and consensus reached of emerging themes (PRJ, CJB, JRC, JBN, JS)
IV	Review of the themes whereby a set of subthemes were explored and refined, including similarities and differences between interviews (PRJ, CJB, JRC, JBN, JS)
V	Themes and subthemes reviewed, revised, and agreed upon, and results summarised and described in the paper (PRJ, CJB, JRC, JBN, JS).

**Table 2 ijerph-18-03924-t002:** Participant characteristics.

General Practice and Geography	Number of Registered Patients	Participants	Gender
Practice no. 1	4000	GP 1.1	M
Rural		GP 1.2	M
		GP 1.3	M
		PN 1.1	F
		PN 1.2	F
Practice no. 2	3000	GP 2.1	M
Urban		PN 2.1	F
Practice no. 3	6000	GP 3.1	M
Urban		GP 3.2	M
		GP 3.3	M
		PN 3.1	F
		PN 3.2	F
Practice no. 4	7500	GP 4.1	F
Rural		GP 4.2	F
		PN 4.1	F

GP = General Practitioner, PN = Practice Nurse, F = Female, M = Male.

**Table 3 ijerph-18-03924-t003:** Identified themes and associated subthemes in relation to attitudes and behaviour towards digital lifestyle interventions in general practice.

Theme 1. Distribution of Roles and Lifestyle Interventions in General Practice	Theme 2. Pros and Cons for Digital Lifestyle Interventions in General Practice
External distribution of roles	Access to real life data
Internal distribution of roles	Change in daily routines

**Table 4 ijerph-18-03924-t004:** Identified factors of importance affecting use of digital individualised coaching of Type 2 diabetes in a general practice setting.

Identified Factors of Importance
1. Needs of knowing the content of the digital lifestyle coaching. What do the GP refer the patient to?
2. Needs of easy overview of real-life data to be able to support self-management and maintenance of healthy lifestyle in patients with T2D
3. Needs of useful feedback from external distributor of lifestyle interventions directly integrated into the electronic health record
4. Needs of less time-consuming consultations and fewer contacts in general practice
5. Unclear distribution and understanding of roles internally and externally in relation to digital lifestyle interventions. Who does what and when?
6. Lack of digital culture and experiences of using eHealth lifestyle interventions in general practice

## Data Availability

Data sharing is not applicable to this article.

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
