# Peer review of "Identification of Important Factors Affecting Use of Digital Individualised Coaching and Treatment of Type 2 Diabetes in General Practice: A Qualitative Feasibility Study"

_ijerph, 2021, doi:10.3390/ijerph18083924_

Round 1
Reviewer 1 Report
Interesting paper, here some remarks, questions
p. 2 l 80, you say that in eight Danish municipalities eHealth lifestyle coaching intervention had been tested and implemented. Q: are the data from these municipalities comparable.
p.2 line 90 ff , these are the research questions, somehow highlight them for easier readiness
distribution of roles and reported time lack - which percentage between other duties have lifestyle issues between the other duties, reality and what would be ideal?
If we have in general no digital culture, how elder people (most of the T2D patients) will feel convinient with eHealth lifestyle interventions, which kind of motivation and incentives can be used?
an eHealth intervention system should be combined with a time management system for the PN and GP
In general the paper lacks from a no use of visualizations - even if the data are qualitative, there are plenty of methods for text visualizations.
Also I think in the proposed interdisciplinary approach the data perspective combined with a data science specialist is missing.
The paper is quite hard to read because it is only text. Also a weighting of the factors of importance for digital lifestyle intervention is missing.
Reviewer 2 Report
This is a potentially useful study, based on having identified two important issues - use of e-health, and acceptability of a specific application type - based on pro-active information seeking. However, the presentation style and development of the overall message are impeded by the narrative analysis reporting, which disrupts the message flow:
- The Themes and Sub-themes lead the presentation of sections of the results. The logic would be better if the example results were presented then the theme distilled out.
- The example in lines 167-169 does not seem to fit under the heading of External Roles - it is a dialogue about different internal actions within the practice.
- A range of other critical success factors, or essential pre-requisites to adoption, are presented which are valuable but appear to be under-valued by the presentation style.
The authors might usefully consider revising the presentation, to focus on critical adoption factors identified from the focus groups, of which understanding of roles would be one element.
Round 2
Reviewer 2 Report
Thank you for significantly improving this paper in the light of feedback.
However, partly as a result of these changes, two smaller questions now emerge:
Lines 88-91 - This insertion is clear, but the implication as a result is that this is a known and validated intervention now being delivered in practices, and with a first consultation on line. This raises other options:
- a core issue is not about diabetes, but about GP and PN coaching, whatever the topic, as the content is not new in the Danish context
- for the first consultation to be in person in the practice
- for the municipal first consultation to be on line
Line 175 - Suggest the recording and transcript be checked to see if "Well" should be "When".
Lines 336-340 - The reference to the Australian study was to the importance of GPs knowing the clinical content of the intervention. But in this case the content was proven and known in municipalities - it is not a new unknown such as an on-line service.
It would be helpful to have these points clarified.
Author Response
Response to reviewer 2:
Thank you for finding our paper significantly improved based on your helpful remarks and suggested revisions. We agree that the smaller questions need to be clarified. Below, we have explained how the individual remarks have been addressed. We have highlighted the changes in our manuscript within the document by using track changes mode in MS Word and by using colored (red) text.
Remark 1:
Lines 88-91 - This insertion is clear, but the implication as a result is that this is a known and validated intervention now being delivered in practices, and with a first consultation on-line. This raises other options:
a core issue is not about diabetes, but about GP and PN coaching, whatever the topic, as the content is not new in the Danish context for the first consultation to be in person in the practice for the municipal first consultation to be on line
Response 1:
Thank you for your question concerning how and by who the coaching is performed. This is now clarified and we have revised the sentence: ”However, in the general practice setting, the first coaching session takes place via video consultations by a trained external online health coach and not in person by a municipality employed health coach like in the municipalities.”
Remark 2:
Line 175 - Suggest the recording and transcript be checked to see if "Well" should be "When".
Response 2:
Sorry for the confusion. It’s a translation from Danish spoken language. We have decided to delete “well” and not replace it with “when” since this will mean something else. “When” would indicate that the general practice does not always receive the report from the municipalities but the fact is that they do but the GP is not aware of this and does not use it like the PN does.
Remark 3:
Lines 336-340 - The reference to the Australian study was to the importance of GPs knowing the clinical content of the intervention. But in this case the content was proven and known in municipalities - it is not a new unknown such as an on-line service.
Response 3:
We agree that the message in this section where we refer to the Australian study is unclear. We have clarified this and hope that it is clearer now. Please look at lines 337-349.
The revised section:
“A recently published Australian study reported that barriers to refer patients to digital interventions were among others the lack of knowledge of effective apps, the lack of a trustworthy source to access them and the lack of understanding of benefits [39]. Our study identified similar important factors that could be a barrier if GPs and PNs should refer the patients to a digital lifestyle intervention. We found that it is important to inform general practices about the digital lifestyle intervention as a valid offer that they can refer their T2D patients to. Additionally, we found that it is important that the GPs and PNs understand the content and the benefits of the digital coaching sessions, and that it is provided by an external healthcare provider they can trust. In addition, it is important that the distribution of roles is clear, so that the patient experience a collaboration between the healthcare providers in primary care and external healthcare providers”.
We hope that you find the revisions to your satisfaction.
